# A multiscale 3D chemotaxis assay reveals bacterial navigation mechanisms

Marianne Grognot [1] & Katja M. Taute [1✉]

How motile bacteria navigate environmental chemical gradients has implications ranging from health to climate science, but the underlying behavioral mechanisms are unknown for most species. The well-studied navigation strategy of *Escherichia coli* forms a powerful paradigm that is widely assumed to translate to other bacterial species. This assumption is rarely tested because of a lack of techniques capable of bridging scales from individual navigation behavior to the resulting population-level chemotactic performance. Here, we present such a multiscale 3D chemotaxis assay by combining high-throughput 3D bacterial tracking with microfluidically created chemical gradients. Large datasets of 3D trajectories yield the statistical power required to assess chemotactic performance at the population level, while simultaneously resolving the underlying 3D navigation behavior for every individual. We demonstrate that surface effects confound typical 2D chemotaxis assays, and reveal that, contrary to previous reports, *Caulobacter crescentus* breaks with the *E. coli* paradigm.

[1] Rowland Institute at Harvard University, Cambridge, MA, USA. ✉email: taute@rowland.harvard.edu

Chemotaxis enables bacteria to navigate external chemical fields and is now recognized as a key factor driving interactions of bacteria with their environment and each other, with wide-ranging effects that include promoting pathogenicity[1], establishing symbioses[2], and shaping geochemical fluxes[3]. Because individual bacterial motility behavior has a large random component, chemotaxis is usually assessed using population-level assays that average over the behavior of thousands of individuals, ranging from Adler's classic capillary assay[4] to modern approaches based on video microscopy[5]. While these approaches are highly sensitive and precise in detecting small chemotactic effects, they are blind to the underlying behavioral mechanisms. Berg's pioneering 3D tracker capable of following a single bacterium swimming in 3D demonstrated the importance of resolving individual 3D motility behavior for revealing chemotactic mechanisms[6]. The resulting understanding that *Escherichia coli* chemotaxis is controlled via the bias in the rotation direction of the flagellum[7] has become a dominant paradigm in bacterial chemotaxis research[8]. Recent findings, however, suggest that many other species, including the majority of marine bacteria, may use a different strategy[9,10], highlighting the need for efficient and broadly applicable methods of characterizing chemotactic behavior.

The pervasive interindividual variability present even in genetically identical populations[11] renders throughput and sampling crucial bottlenecks in characterizing chemotactic behavior based on trajectory data. Methods aimed at revealing potentially diverse chemotactic mechanisms must bridge the scales between individuals and populations by capturing motility behavior of individual bacteria with the statistical power to simultaneously reveal chemotactic performance, which is typically only accessible via ensemble averages.

Here we introduce a chemotaxis assay that enables such a multiscale approach by harnessing a recently developed high-throughput 3D tracking method[12] to capture individual navigation behavior in the presence of microfluidically created chemical gradients for thousands of bacteria within minutes. After validating our approach using the well-characterized *E. coli* model system, we demonstrate its power by revealing that the chemotactic mechanism of the freshwater bacterium *Caulobacter crescentus* breaks with the *E. coli* paradigm.

## Results

**Multiscale assay accurately quantifies *E. coli* chemotaxis.** In our assay, typically 50–100 bacteria are tracked simultaneously in 3D in the center of a quasi-static, linear gradient field created in a microfluidic device (Fig. 1a, Supplementary Fig. 1, and Methods). Typically, thousands of trajectories are gathered in minutes, enabling a precise determination of the chemotactic drift velocity, $v_d$, as the population-averaged velocity along the direction of the gradient ($x$) (Fig. 1b and Supplementary Table 1). We demonstrate the technique by assessing chemotaxis of the well-characterized *E. coli* strain AW405 towards the non-metabolizable chemoattractant α-methyl-DL-aspartate (MeAsp, Fig. 1b–e and Supplementary Fig. 1e). The population-averaged drift velocity of $2.7 \pm 0.3\ \mu m\,s^{-1}$ (mean ± SD across three biological replicates) for bulk trajectories in a linear gradient of $10\ \mu M$ $mm^{-1}$ MeAsp aligns well with values imputed from previous work (Supplementary Discussion), confirming that our technique provides an accurate quantification of chemotactic performance.

***E. coli*'s chemotactic drift decreases near the chamber surface.** Resolving the drift velocity, $v_d$, as a function of vertical position, $z$, reveals that the drift velocity is roughly constant in the bulk liquid, but decreases sharply near the surfaces of the sample

chamber (Fig. 1d and Supplementary Fig. 1). We attribute this decrease to trajectory curvature at the surfaces (Fig. 1c) randomizing bacterial orientations and thus leading to a decreased chemotactic response. Such curvature results from hydrodynamic interactions with the surface[13] and is thus present in common 2D motility assays that increase trajectory durations by either constraining the bacteria to a thin sample chamber[14] or by limiting observations to a chamber surface[15]. Our findings indicate that many standard 2D chemotaxis assays may be confounded by surface effects in their ability to quantify chemotactic performance, demonstrating the value of 3D tracking.

**Multiscale assay captures *C. crescentus* chemotaxis and reveals phenotypically distinct swarmer cell populations.** To demonstrate the power of our approach for revealing novel chemotactic behaviors, we turn to the freshwater bacterium *C. crescentus* whose weak, cell cycle-dependent chemotaxis response[16,17] has been challenging to capture. In *C. crescentus*'s life cycle, cell division occurs in stalked sessile cells and produces swarmer cells whose "run-reverse-flick" motility[9,18] is driven by a single polar flagellum. Either direction of flagellar rotation results in locomotion, with the flagellum either pushing the cell body forward or pulling it backward. Reversals (turns by ~180°) occur with the transition from pushing to pulling, whereas the opposite transition is accompanied by a so-called flick, an approximately right-angle turn (Fig. 2a).

We recorded 79,244 individual 3D bulk trajectories of motile *C. crescentus* swarmer cells navigating a $1\ mM\,mm^{-1}$ xylose gradient in five biologically independent experiments (Supplementary Table 2). To limit differences in cell cycle state, the observed swarmer cells were grown from stalked cells on the sample chamber surfaces during the experiment (Methods). Strikingly, 54% of the more than 123,000 s of total trajectory time we obtained consists of straight trajectories with no turns and no discernible chemotactic drift up the gradient. A statistical analysis of individual turn event frequencies supports the notion that these "smooth swimmers" form a phenotypically distinct group (Supplementary Discussion). In *C. crescentus*'s life cycle, the swarmer-to-sessile cell transition is accompanied by intracellular biochemical changes that may favor smooth swimming, including the degradation of chemoreceptors[17] and a rise in c-di-GMP[19] (Supplementary Discussion). While we do not know the origin of the smooth-swimming population, one possibility is that it represents the early stages of the swarmer-to-sessile cell transition. Smooth swimming is likely to increase both the rate of surface encounters as well as the time spent swimming along the surface and may be an integral part of *C. crescentus*'s strategy for completing the swarmer-to-sessile cell transition[19].

The drift up the gradient, $v_d$, of the remaining, turning population amounts to only $0.26 \pm 0.12\ \mu m\,s^{-1}$ (mean ± SEM), corresponding to less than 0.5% of their average swimming speed, $v$, of $56\ \mu m\,s^{-1}$. The speed of backward runs, with the flagellum pulling the cell, is $2.5 \pm 0.2\%$ higher than that of forward ones (Supplementary Fig. 2). Forward and backward run duration distributions are approximated well by inverse Gaussian distributions (Supplementary Fig. 2), consistent with previous reports[15,20].

***C. crescentus* performs chemotaxis at near-constant bias.** *C. crescentus* chemotaxis has been assumed to follow the *E. coli* paradigm, where the cytoplasmic concentration of phosphorylated CheY ([CheY-P]), the chemotaxis response regulator, modulates the fraction of the time that the flagella rotate clockwise (CW), the so-called CW bias[19,20]. In *E. coli*, counterclockwise (CCW) rotation supports locomotion ("runs"), whereas

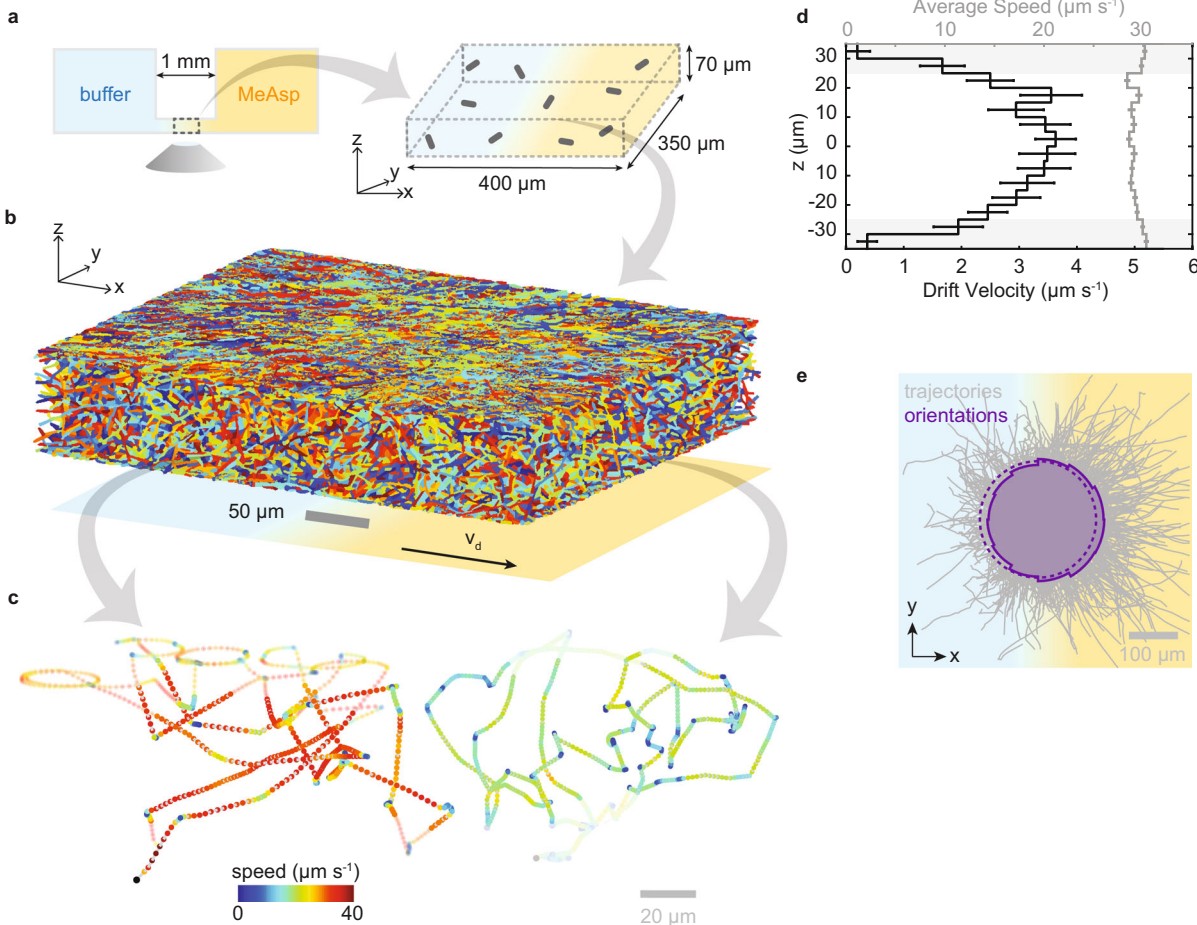

**Fig. 1 Schematic of multiscale chemotaxis assay and its application to *E. coli* strain AW405. a** A quasi-static linear chemical gradient is established between two reservoirs containing a uniform concentration of bacteria. Bacteria are observed in the central portion of the linear gradient starting 50 min after filling the reservoirs. **b** 5045 individual trajectories with a minimum duration of five frames and containing 37,080 s of total trajectory time, obtained in 9 min of recording at 15 Hz in a typical experiment. **c** Two example trajectories (durations 63 and 65 s) showing run-tumble motility in bulk solution and circular segments near the chamber surface (within 10 μm distance, faded). **d** Drift velocity (black, defined as the average speed along the gradient direction, *x*) and average swimming speed (gray) as a function of height, *z*, computed from three biological replicates of the experiment, comprising 9903 motile trajectories with a combined duration of 79,562 s. Only bulk trajectories (defined as trajectory segments with a distance of more than 10 μm to the surface) are retained for further analysis. Error bars reflect standard errors of the mean. **e** Bulk trajectories from the same dataset with aligned origins (gray) and polar probability distribution of instantaneous swimming directions projected in the *x-y* plane (purple, solid line). A flat distribution (dashed) is shown for reference. For visual clarity, only those 7688 trajectories with a minimal duration of 1 s are shown, comprising 24,755 s of trajectory data. The polar distribution of orientations is based on the full dataset (9294 trajectories with duration 26,329 s).

CW rotation induces reorientation events ("tumbles") (Fig. 2b). Chemotaxis is achieved by dynamic modification of the bias so as to increase the duration of runs aligned with the gradient direction. Early studies hypothesized that in *C. crescentus*, forward runs driven by CW rotation correspond to *E. coli* runs, while backward runs driven by CCW rotation are equivalent to *E. coli* tumbles because of their shorter duration[21,22]. Given similar forward and backward swimming speeds, a bias towards CW rotation would then yield net displacements in the forward swimming direction. In line with this hypothesis, a recent study of 2D surface swimming behavior in oxygen gradients found that forward, but not backward runs, when directed up the gradient compared to down[15].

In contrast with this hypothesis, we find that both forward and backward run segments are extended when ascending, versus descending, a chemoattractant gradient (Fig. 2c and Supplementary Fig. 2). This ranking is reproduced independently in four of five individual replicate experiments combined here, though at lower significance (Supplementary Fig. 3). A fifth recording shows

similar backward run durations ascending and descending the gradient, but the error bars allow either ranking.

In fact, our data support a constant motor bias, that is, a constant ratio of forward versus backward swimming interval durations (Fig. 2f) and thus indicate a radical break with the *E. coli* paradigm of motor bias-driven chemotaxis in *C. crescentus*. We propose that, in *C. crescentus*, [CheY-P] lowers the energy barrier between the two rotation states, but leaves the states' relative energy levels unchanged, thus modulating switching rates without affecting the motor bias[10] (Fig. 2d and Supplementary Discussion). This interpretation is also consistent with the puzzling previous observation that, in sharp contrast to *E. coli*[11], *C. crescentus* has been reported to show hardly any variability in motor bias between individuals[20], suggesting that its motor bias might be unaffected by cytoplasmic fluctuations in [CheY-P].

**Competition between chemotaxis and diffusion.** Smooth swimmers surprisingly show a down-gradient drift of -(0.8 ± 0.2)

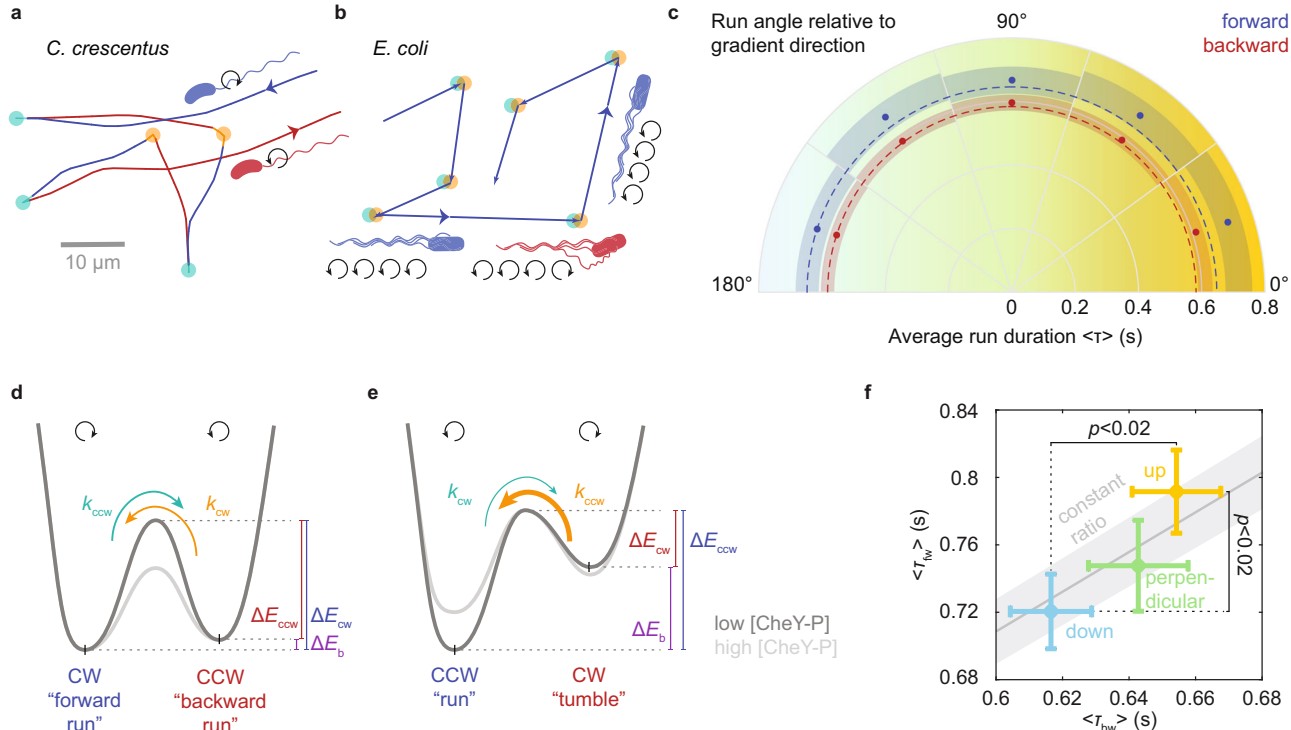

**Fig. 2 C. crescentus chemotaxis. a** Example trajectory showing alternating backward (red) and forward (blue) runs, separated by switches in flagellar rotation direction that result in reversals (CW to CCW, teal) or flicks (CCW to CW, orange). **b** Schematic of *E. coli* run-tumble motility. Runs are driven by CCW rotation (blue). Tumbles result from temporary CW rotation (red) of at least one flagellum and thus are bordered by two switches in rotation direction (CCW to CW, teal, CW to CCW, orange). **c** Radial plot of average run durations (blue: forward, red: backward runs) as a function of projected angle to the *x*-axis in the *x-y* plane. The dashed lines indicate the average run durations observed for swimming down the gradient and serve to facilitate comparison. Shading indicates 95% confidence intervals. Values are computed from a total of 2898 forward and 5342 backward runs. **d** Schematic of two-state motor rotation model proposed for *C. crescentus* and **e** established for *E. coli*[38, 39]. The energy difference $\Delta E_b$ (purple) between states determines the motor bias, while the energy barrier between states, $\Delta E_{ccw}$ and $\Delta E_{cw}$, determines the switching rates, $k_{cw}$ and $k_{ccw}$, respectively (see Supplementary Discussion). **f** Average forward versus backward run durations, $<\tau_{fw}>$ versus $<\tau_{bw}>$, up (yellow), down (cyan), or perpendicular to (green) the gradient (defined by 36° cones around positive *x*-axis, negative *x*-axis, or *y*-axis, respectively). The solid line reflects a best-fit constant CW bias of 0.54 ± 0.01, with the standard error (Methods) shown as gray shading. *P* values are shown for one-sided *t*-tests between durations up and down the gradient. Error bars reflect standard error of the mean. Averages and *p* values are determined over 506, 449, 408 forward runs and over 882, 946, 835 backward runs ascending (yellow), descending (cyan), and perpendicular to (green) the gradient, respectively.

µm s$^{-1}$ (mean ± SEM). We attribute this drift to a diffusive flux driven by an imbalance in bacterial concentration between the reservoirs. Automated growth measurements show a growth rate advantage in the presence of xylose, and we find that the bacterial density of both smooth-swimming and turning cells is ~80% higher in the xylose-containing reservoir than in the other one (Supplementary Fig. 4). A theoretical prediction of the expected down-gradient diffusive drift based on the observed density difference and an estimated effective diffusion coefficient yields an estimate of $v_{diff} = -0.6$ µm s$^{-1}$ for smooth swimmers, in good agreement with the observed value of -(0.8 ± 0.2) µm s$^{-1}$ (Supplementary Discussion).

The up-gradient drift velocity, $v_d$, observed for turning bacteria may thus represent a competition between chemotaxis and diffusive drift: $v_d = v_d^* + v_{diff}$, where $v_d^*$ represents the drift expected solely from chemotaxis. This chemotactic contribution can be estimated separately based on the experimentally obtained orientation-dependent average run durations, $<\tau(\theta)>$:

$$v_d^* = v_0 \frac{\int_0^\pi \langle \tau_{fw}(\theta) \rangle \cos\theta \sin\theta\, d\theta + \int_0^\pi \langle \tau_{bw}(\theta) \rangle \cos\theta \sin\theta\, d\theta}{\int_0^\pi \langle \tau_{fw}(\theta) \rangle \sin\theta\, d\theta + \int_0^\pi \langle \tau_{bw}(\theta) \rangle \sin\theta\, d\theta},$$
(1)

where $\theta$ is the angle between the run direction and the gradient direction. As the average swimming speed only varies by a

standard deviation of about 14% between motile individuals, we neglect any potential correlation between swimming speed and run duration modulation and base our estimate on the average swimming speed, $v_0$. We thus also neglect the small difference between forward and backward run speeds. With $v_0 = 54$ µm s$^{-1}$, we then obtain $v_d^* = (0.43 ± 0.21)$ µm s$^{-1}$ (mean ± SEM) as the estimated chemotactic drift velocity from the measured run duration modulation (see Methods for details).

While this value is within error of the measured drift velocity of $v_d = (0.26 ± 0.12)$ µm s$^{-1}$, their difference $v_d - v_d^* \approx -(0.17 ± 0.33)$ µm s$^{-1}$ is also consistent with a theoretical prediction of $-0.2$ µm s$^{-1}$ for the diffusive down-gradient drift of the turning population based on the observed density imbalance (Supplementary Discussion). We thus conclude that the experimentally observed drift velocity for *C. crescentus* likely slightly underestimates the chemotactic drift velocity due to a competition between chemotaxis and effective diffusion.

## Discussion

*C. crescentus* has been shown to demonstrate strong taxis to oxygen[15]. Such so-called aerotaxis is usually differentiated from chemotaxis because it is thought to be driven by indirect intracellular sensing, rather than transduction of extracellular chemoattractant ligand binding by chemoreceptors[23]. While several

chemoattractants for *C. crescentus* have been reported in the literature[21,24,25], the fact that no similarly strong response to any of them has been demonstrated suggests that chemotaxis may be either generally weak in *C. crescentus* or may be reserved to conditions or subpopulations that are challenging to capture experimentally.

While multiple sugars have previously been reported to be chemoattractants for *C. crescentus* on the basis of soft agar plate assays[21,24,25], these assays can be ambiguous to interpret[26]. We chose xylose because, to our knowledge, it is the only compound for which a response at the level of individual swimming behavior has been reported[24]. We cannot exclude that the presence of glucose in our assay lowers the observed chemotactic response to xylose. Such an effect would be expected if xylose and glucose were sensed by the same receptor, as is the case for *E. coli*[27]. The chemoattractant ligand repertoires of the 18 putative chemoreceptors encoded in the *C. crescentus* genome[28], however, have not been characterized. Going forward, our assay presents the opportunity to identify and validate further chemoattractants.

Many chemoattractants are nutrients and confer a growth benefit[29]. Competition between chemotaxis and diffusive drift driven by the resulting inhomogeneous growth is thus expected to also be present in natural settings where bacteria use chemotaxis to find and exploit nutrients. Both behavioral components can be replicated and determined in our assay. In the present experiment, the density imbalance is exacerbated by the fact that observed cells are created by growth and release from surface-attached cells during the experiment. In the more typical scenario applicable to most other species, where a bacterial suspension is flowed into the device and inspected once the gradient is stable, any growth-conferred density imbalances between the two reservoirs are expected to remain much smaller (Supplementary Discussion). In addition, non-metabolizable chemoattractants such as the MeAsp used in the *E. coli* experiments here can be used to prevent growth-driven density imbalances altogether.

While the magnitude of the drift velocity measured for *C. crescentus* is small and may slightly underestimate the chemotactic drift due to a diffusive counter flux, we can, however, confidently assign its origin to chemotaxis based on the observed modulation in run durations with respect to the orientation relative to the gradient (Fig. 2c, f). This modulation is reproduced independently in each of the five replicate experiments that were pooled to compute the drift velocity (Supplementary Fig. 3). Our conclusion that *C. crescentus*'s chemotaxis strategy deviates from that of *E. coli* does not rely on the exact magnitude of the observed response.

The sensitivity limits of our assay are dominated by statistical requirements. The confidence in a measured drift value, $v_d$, depends on the ratio, $f = SE(v_d)/|v_d|$, of its error estimate and its absolute value. Based on purely statistical uncertainties, the ratio, $f$, that can be achieved with a given number of trajectories, $N$, can be estimated as

$$f = \frac{v_0}{\sqrt{3N}|v_d|} \qquad (2)$$

(Supplementary Discussion). Our values for a typical *E. coli* experiment, $N = 3 \times 10^3$, $v_d = 2.7 \ \mu m \ s^{-1}$, and $v_0 = 30 \ \mu m \ s^{-1}$, predict $f = 0.12$, in good agreement with the ~10% variation in results between repeats of the experiment (Supplementary Table 1). For the *C. crescentus* data, $v_0/v_d$ is more than an order of magnitude larger than for *E. coli*, and we pool multiple experiments into one dataset containing ~24 times as many trajectories as a typical *E. coli* experiment. The predicted value of $f = 0.43$ agrees well with the experimental estimate of $f \approx 0.46$ in *C. crescentus*. Statistical errors thus account for the sensitivity limits of our assay.

While, in principle, arbitrary levels of precision can be reached by increasing the amount of data gathered, throughput becomes a key determinant of the practical feasibility of such an approach. The detection of even the small chemotactic drift in *C. crescentus* highlights the sensitivity enabled by the high throughput of our technique. The repeatability of our *E. coli* drift velocities demonstrates that, under typical conditions, our assay enables the quantitative characterization of chemotactic performance from routine, individual experiments.

In contrast to our finding that *C. crescentus* modulates both forward and backward run durations, previous work indicates an *E. coli*-like modulation of only the forward run durations[15]. The apparent conflict likely results from a technical artifact imposed by the constraints of 2D tracking. To increase the typical time a bacterium spends in the focal plane, 2D bacterial tracking had been performed at the sample chamber surface[15] which hydrodynamically attracts the bacteria[30]. The surface-induced trajectory curvature is more pronounced in the backward than in the forward runs[15,31] (Supplementary Fig. 2g), thus likely diminishing the chemotactic response more strongly for backward than for forward runs. Placing the focal plane in the bulk can prevent such surface effects in 2D tracking but incurs severely shortened trajectories that are less likely to fully capture runs whose orientation can be assigned based on their bordering turning events. We estimate that, under typical 2D tracking conditions, ~30–40 times as much data would need to be obtained to detect the dependence of run durations on orientation relative to the gradient at a similar fidelity (Supplementary Discussion and Supplementary Fig. 5). This example, together with the finding of a decreased drift velocity for *E. coli* close to surfaces, highlights the crucial significance of full 3D behavioral information when assessing chemotactic mechanisms: 3D tracking enables the acquisition of long trajectories without a need for bacterial confinement as well as accurate turning angle measurements for determining bacterial orientation even for short trajectories with few turning events.

The mechanism we unveil for the alpha proteobacterium *C. crescentus* aligns with recent findings for the singly flagellated gamma proteobacteria *Vibrio alginolyticus*[9] and *Pseudomonas aeruginosa*[10]. These species also extend both forward and backward swimming intervals during chemotaxis, suggesting that this mechanism may be much more common than previously assumed. To our knowledge, no species with polar flagella has conclusively been shown to follow the *E. coli* scheme, raising the intriguing possibility that the influential *E. coli* paradigm may reflect a special case limited to species that share its flagellation pattern. We note that, despite the stark contrast in motor switching schemes, the resulting swimming pattern suggests a unified behavioral paradigm: any run in a favorable direction is extended.

In summary, our multiscale technique offers simultaneous access to individual and population-level 3D motility behavior and is poised to offer key insights into novel chemotactic mechanisms as well as into the effects of phenotypic heterogeneity on population-level motility behaviors. In contrast to many flow-based chemotaxis assays[32] that are limited to liquid environments, our assay is also compatible with environments such as hydrogels that more closely mimic the complexities of many natural habitats, and thus paves the way for studies of chemotactic mechanisms in ecologically relevant settings.

## Methods

**Microfluidic device and gradient stability**. Quasi-static chemical gradients are created in a commercially available microfluidic device (IBIDI μ-slide Chemotaxis) featuring an ~70 μm high, 1 mm long, and 2 mm wide channel connecting two 65 μl reservoirs. Gradient establishment and stability over time were characterized for fluorescein gradients (10 μM mm$^{-1}$ in MotM, see Table 1 for media compositions) using a confocal microscope (Zeiss AXIO Imager.Z2). The 488 nm line was focused through a 20x water immersion objective and fluorescence emission collected in the

**Table 1 Growth and motility media used.**

| | |
|---|---|
| MotM | 10 mM KPO$_4$ |
| | 0.1 mM EDTA |
| | 1 µM L-methionine |
| | 10 mM lactic acid |
| | 67 mM NaCl |
| | pH 7.0 |
| M2G | 12.3 mM Na$_2$HPO$_4$ |
| | 7.8 mM KH$_2$PO$_4$ |
| | 9.3 mM NH$_4$Cl |
| | 0.5 mM MgSO$_4$ |
| | 0.5 mM CaCl$_2$ |
| | 10 µM FeSO$_4$.7H$_2$O in 8 µM EDTA |
| | 0.2% glucose |
| | pH 6.8 |
| TB | 1% Bacto Tryptone |
| | 0.5% NaCl |
| | pH 7.0 |
| PYE | 0.2% Bacto Peptone |
| | 0.1% Bacto Yeast extract |
| | 1 mM MgSO$_4$ |
| | 0.5 mM CaCl$_2$ |
| | pH 7.0 |

500–585 nm window with a pinhole adjusted to 30 µm diameter. A 4.5 h time series of line scans in the gradient direction across the center of the device was acquired after closing the device (Supplementary Fig. 1b). A large-scale 4 × 4 mm view of the device was obtained by tiling 2D scans (Supplementary Fig. 1a) acquired 5 h after closing the device. The gradient is established within minutes of filling the device and shows a deviation of less than 4% from the final plateau value after 30 min (Supplementary Fig. 1c). No detectable variation in relative gradient magnitude is observed in the time range from 40 min to 4.5 h after filling the device (Supplementary Fig. 1c).

Diffusive timescales scale with the inverse of the diffusion coefficient, and the diffusion coefficient of fluorescein[33] is about half that of aspartate[34]. An aspartate gradient would thus establish even more rapidly than a fluorescein gradient and be stable about half as long as a fluorescein gradient. The stability of a fluorescein gradient over 4.5 h thus implies the stability of an aspartate gradient over 2 h. Our measurements are conducted within 1 h, well within the limits of stability of the chemical gradient.

The chemotactic drift of bacteria up the gradient is not sufficient to cause a substantial imbalance in bacterial concentrations between the two reservoirs. Assuming a uniform drift speed of ~3 µm s$^{-1}$ along the 1 mm long channel, ~3600 s × 3 µm s$^{-1}$/(1000 µm) ≈11 channel volumes of bacteria are transported up the gradient in 1 h. The volume of the channel is $V_c$ = 1 mm × 1 mm × 0.07 mm = 0.07 µl, much smaller than the 65 µl volumes of the reservoirs. In 1 h, the bacterial concentration of the reservoirs thus would change by at most (11 × 0.07 µl)/65 µl ≈ 1.2% as a result of chemotactic drift.

## E. coli experiments
*Bacterial culturing.* Overnight cultures were inoculated from a frozen glycerol stock of *E. coli* AW405 (a kind gift of Howard Berg) in 2 ml TB and grown to saturation at 30 °C, 250 rpm. Day cultures were inoculated with the overnight cultures at 1:200 dilution in 10 ml TB and grown at 33.5 °C, 250 rpm, until they reached an optical density (OD) between 0.3 and 0.35 at 600 nm. Volumes of 1 ml of bacterial culture were washed by three rounds of centrifugation in 1.5 ml microcentrifuge tubes (6 min at 2000 rcf), each followed by gentle resuspension in 1 ml of motility medium MotM. They were diluted to a target OD of 0.003 (for acquisitions in a gradient) or 0.005 (no gradient) in MotM supplemented with 0.002% Tween 20, with or without chemoattractant, for injection into the chemotaxis device.

*Sample preparation.* The device's reservoirs were filled with the two bacterial solutions (with or without chemoattractant) following a modified version of the manufacturer's "Fast Method" protocol. First, the entire device was overfilled with buffer free of chemoattractant or bacteria through the filling ports, and then the central channel's ports were closed with plugs. About 65 µl was removed from one reservoir, replaced by 65 µl of chemoattractant-free bacterial solution, and then this reservoir's ports were closed. Finally, all liquid was removed from the other reservoir and replaced with bacterial solution containing chemoattractant. Key to reproducible gradients is to not overfill this reservoir to avoid liquid flow in the central channel when the last two ports are closed. A uniform bacterial density across the device ensures that any population drift observed is not the result of a diffusive flux, but likely indicates chemotaxis. For control measurements, neither

bacterial solution contained chemoattractant. At the bacterial densities used (OD of 0.005 or less), oxygen depletion is unlikely, and we do not observe a change in *E. coli* swimming speed in the reservoirs over the course of 1 h (Supplementary Fig. 1d).

*Data acquisition.* Phase contrast microscopy recordings were obtained at room temperature (~22 °C) on a Nikon Ti-E inverted microscope using an sCMOS camera (PCO Edge 4.2) and a 40x objective lens (Nikon CFI SPlan Fluor ELWD 40x ADM Ph2, correction collar set to 1.2 mm to induce spherical aberrations[12]) focused at the center of the channel in all three dimensions. 3D bacterial trajectories were extracted[12] for a field of view of ~350 µm × 300 µm laterally (*x, y*) and over the entire depth (*z*) of the channel for typically several dozen individuals at a time. For *E. coli*, recordings were obtained starting from 50 min after filling the device. Three 3-min long recordings were obtained at 15 fps. Three biological replicates can be performed in parallel in half a day.

*In-device conditions.* We confirmed that the *E. coli* motile population we tracked in the central channel was representative of the whole population by also acquiring trajectories in one reservoir during an experiment. The average speed of the motile population, defined as individuals having a mean swimming speed larger than 10 µm s$^{-1}$, in the reservoirs was stable in time and similar to that observed in the central channel (Supplementary Fig. 1d).

*Data analysis.* 3D Trajectories were extracted from phase contrast recordings using a high-throughput 3D tracking method based on image similarity between bacteria and a reference library[12]. One individual may, in principle, contribute more than one trajectory, either because it leaves and reenters the tracking volume or because the tracking algorithm briefly loses it and then finds it again. Trajectories shorter than five frames were discarded. Positions were smoothed using second order ADMM-based trend-filtering[35] with regularization parameter $\lambda = 1$, and speeds computed as forward differences in positions divided by the time interval between frames. All trajectories with an average speed below a threshold were considered non-motile and discarded. The threshold was set at 15 µm s$^{-1}$ unless noted otherwise. The *z* position of the top and bottom of the chamber were identified by visual inspection of trajectory data. All trajectory segments within 10 µm of the top or bottom of the central channel were removed to avoid surface interaction effects, retaining 35% of total trajectory time. The drift velocity is the average of the *x* component of all 3D speed vectors from all bacteria. Across three biological replicates performed in parallel, we obtain a drift velocity of 2.7 ± 0.3 µm s$^{-1}$ (mean ± SD across the replicates) for *E. coli* in a 10 µM mm$^{-1}$ MeAsp gradient and observe no chemotactic drift along either the *y* or *z* axis (0.1 ± 0.3 µm s$^{-1}$ and 0.09 ± 0.1 µm s$^{-1}$, respectively). Biological replicates performed on other days yielded drift velocities of 2.7 and 2.5 µm s$^{-1}$. A control chamber without a gradient showed no drift either (0.34 ± 0.8 µm s$^{-1}$ along *x*). For data obtained from a single experiment (Supplementary Fig. 1e), we estimate the noise on the drift measurement by a jackknife resampling procedure consisting of dividing the data into subsets of 150 trajectories and computing the standard error of the mean drift obtained for different subsets. For drift as a function of *z*, trajectories are first sliced into segments by *z* bin, and jackknifing is performed for each *z* bin. Statistical descriptors of all *E. coli* datasets are summarized in Supplementary Table 1.

## C. crescentus experiments
*Bacterial culturing and sample preparation.* Overnight cultures were inoculated from individual *C. crescentus* (CB15, ATCC 19089) colonies, grown on 1.5% agar PYE plates streaked from glycerol stock, and grown to saturation in 2 ml PYE at 30 °C, 200 rpm. Day cultures were inoculated at a dilution of 1:20 (v/v) in M2G[36] or PYE and grown to an OD600 of at least 0.3 (a few hours for PYE, around 12 h for M2G). Because the cell cycle-dependent motility of *C. crescentus* is quickly lost, we opted to grow the cells directly inside the device and let them produce swarmer cells while the gradient is being established. To this end, the day culture was again diluted 1:1 in fresh medium and injected into both reservoirs of the device which was then incubated at room temperature. When the chamber walls were colonized by a sufficient density of stalked cells as determined by visual inspection under the microscope (a few hours for PYE, 2 days for M2G), the device was rinsed several times with fresh M2G, until no swimming bacteria were observed. Then fresh M2G and 1 mM xylose/M2G, respectively, were injected into the reservoirs to create a 1 mM mm$^{-1}$ xylose gradient in the central channel. Xylose is a known chemoattractant for *C. crescentus*[21,24].

*Data acquisition.* Because of the cell cycle-dependent chemotaxis[16] of *C. crescentus*, we favored acquiring data as early as possible over waiting for perfect gradient stability. Recordings spanning 2.5 min at 30 fps were acquired from 20 to 55 min after closing the device, in five biologically independent experiments, totaling a cumulated acquisition time of 75 min.

*Data analysis.* Trajectories were obtained and smoothed as for *E. coli*, except for the ADMM regularization parameter being set to $\lambda = 0.3$. To account for the layer of attached cells lining the surface, only segments with a distance of more than 13 µm from the top or bottom chamber surface were retained to avoid surface interaction

effects. Supplementary Table 2 details statistical characteristics of the subsets of data used for analysis. Standard errors on drift velocities are obtained by jack-knifing as described for *E. coli*.

*Run-reverse-flick analysis.* The turning event detection is based on the local rate of angular change, computed from the dot product between the sums of the two consecutive velocity vectors preceding and subsequent to a time point. The threshold for a turn to begin is an α-fold rate relative to the median rate of angular change of the run segments, as determined in three iterations of the procedure. We determined by visual inspection of trajectories that a factor α = 6 gave satisfactory results. A new run begins with at least two time points (at least 0.066 s) under this threshold. Backward (CCW rotation) and forward (CW rotation) runs were identified as runs with a turn under 130°, respectively at the end or at the beginning the run, and a turn above 150° at the other end of the run. A total of 5342 backward and 2898 forward runs were identified within a subpopulation of 6230 trajectories.

*Run duration analysis.* Runs are considered to be going up or down the gradient if they fall within a 36° cone around the positive, respectively negative, x-axis. The value of 36° was chosen so as to balance a trade-off between maximizing the number of contributing runs with maximizing their alignment with the gradient direction. The conclusions are not sensitive to the exact value chosen. The fact that runs going up the gradient are longer than those going down the gradient confirms that the observed drift, though small, is indeed caused by chemotaxis. We determine maximum-likelihood inverse Gaussian distributions from the run duration data:

$$f = \sqrt{\frac{\lambda}{2\pi\tau^3}} \exp\left(-\frac{\lambda(\tau - \mu)^2}{2\mu^2\tau}\right)$$

where the run duration $\tau$, the mean, $\mu$, and the shape parameters, $\lambda$, are strictly positive. We obtain the following parameters: $\mu = 0.79$ s, $\lambda = 1.18$ s, and $\mu = 0.72$ s, $\lambda = 1.21$ s, for forward runs going up and down the gradient, respectively; and $\mu = 0.65$ s, $\lambda = 1.85$ s, and $\mu = 0.62$ s, $\lambda = 1.78$ s, for backward runs going up or down the gradient, respectively (Supplementary Fig. 2f).

*Motor bias analysis.* To determine a best-fit motor bias for Fig. 2f, an orthonormal linear fit constrained to a zero intercept is applied to the average run durations up, down, and perpendicular to the gradient. The error on the slope is estimated as the standard deviation of slopes obtained by fitting data generated in a Monte Carlo procedure, consisting of drawing data points randomly from Gaussian distributions centered about the actual data points and with a standard deviation matching the data points' standard error. The CW bias, $b_{CW}$, can be obtained from the slope, $s$, as $b_{CW} = s/(1+s)$. We obtain $s = 1.18 \pm 0.03$, corresponding to $b_{CW} = 0.54 \pm 0.01$.

*Run speed analysis.* To determine the ratio of forward to backward swimming speed for Supplementary Fig. 2e, average speeds were computed across all forward and backward runs, respectively, for each trajectory in subset 8 (see Supplementary Table 2). An orthonormal linear fit constrained to zero intercept yields a slope of $1.025 \pm 0.002$ (SE).

*Chemotactic drift velocity.* The chemotactic contribution to the drift velocity, $v_d^*$, can be estimated on the basis of the measured orientation-dependent average run durations:

$$v_d^* = \frac{v_0\left(\int_0^\pi \langle \tau_{fw}(\theta)\rangle \cos\theta \sin\theta d\theta + \int_0^\pi \langle \tau_{bw}(\theta)\rangle \cos\theta \sin\theta d\theta\right)}{\int_0^\pi \langle \tau_{fw}(\theta)\rangle \sin\theta d\theta + \int_0^\pi \langle \tau_{bw}(\theta)\rangle \sin\theta d\theta}$$

where $\theta$ is the angle between the run direction and the gradient direction. The $\sin\theta$ term represents the frequency of runs with orientation $\theta$ under the assumption that all orientations in 3D are equally likely. The denominator is proportional to the average duration of runs, while the numerator is proportional to the average spatial displacement during a run along the gradient direction. Forward and backward runs are taken to be equally likely as they alternate.

In practice, we estimate $\langle\tau(\theta)\rangle$ as the average duration of runs with a maximum elevation of 36° to the x-y plane and an angle $\theta$ between the x-y projection of the run direction and the gradient direction, which is the positive x-axis. Averages are computed separately for five bins of width 36° in $\theta$. We thus approximate $v_d^*$ as

$$v_d^* \approx v_0 \frac{\sum_{i=1}^5 \langle\tau_{fw}^{ij}\rangle \cos\theta_i \sin\theta_i + \sum_{i=1}^5 \langle\tau_{bw}^{ij}\rangle \cos\theta_i \sin\theta_i}{\sum_{i=1}^5 \langle\tau_{fw}^{ij}\rangle \sin\theta_i + \sum_{i=1}^5 \langle\tau_{bw}^{ij}\rangle \sin\theta_i},$$

where $\theta_i$ marks the center of the $i^{th}$ bin. Run durations, $\tau^{ij}$, are averaged over all runs ij with an orientation $\theta_{ij}$ that falls into the $i^{th}$ bin. With $v_0 = 54$ µm s$^{-1}$, we then obtain $v_d^* = (0.43 \pm 0.21)$ µm s$^{-1}$ (mean ± SEM) as the estimated chemotactic drift velocity from the measured run duration modulation. The error is estimated in a jackknifing procedure that divides the data into random subsets of 500 trajectories each. An estimate of $v_d^*$ is computed as above from the runs contained in each subset, and the error is determined as the standard error of the mean of these estimates.

*Imbalance in bacterial concentration between reservoirs.* In a device prepared as for a regular chemotaxis experiment, we acquired recordings in the middle of the reservoirs instead of the central channel. A time series of 50 s-long recordings was obtained in the xylose-containing reservoir from a few minutes to ~50 min after closing the device. Then, three such recordings were obtained for the center of the other reservoir. Fifty-three minutes after closing the device, the content of each reservoir was retrieved and diluted by threefold in fresh M2G or 1 mM xylose/M2G, respectively. The solutions were flowed into sample chambers, consisting of three layers of parafilm as spacers between a microscopy slide and a #1 coverslip that were heated and pressed to seal. After filling, the ends of the filled chamber were sealed with molten valap (a mixture of vaseline, lanolin, and paraffin) and immediately brought to the microscope for one 100 s recording each, both within 7 min of retrieval from the reservoirs. For density determination, only motile trajectories contained within a z-range of 5–90 µm and with a minimum duration of 0.8 s were included. The z range restriction serves to limit the effect of localization errors caused by light scattering in the reservoirs and by small particulate debris in the retrieved solutions.

*Growth rate experiment.* The growth advantage conferred by xylose was determined in an automated growth experiment in a Multiskan™ FC microplate photometer (Thermo Scientific, software SkanIt 4.1). The wells of a 96-well microplate were filled with 200 µl of either PYE alone (blank wells and border wells) or with a saturated culture diluted in fresh PYE. After 24 h incubation at room temperature, the wells were washed by removing liquid and refilling with fresh M2G medium. After further 24 h of incubation at room temperature, the wells were washed three times as above and then filled with either 200 µl M2G (blanks, borders, and two rows of 10 wells each) or M2G + 1 mM Xylose (two rows of 10 wells each). A BreathEasy$^R$ sealing membrane (Sigma-Aldrich) was placed on top of the 96-well microplate. Then the microplate was placed in the microplate reader and incubated at 30 °C with low continuous shaking. Absorbance at 620 nm was recorded every 5 min for the first 3 h and then every 15 min for another 10 h.

**Statistics and reproducibility**. Bacterial trajectories sample sizes were chosen to balance reasonable acquisition time, chemoattractant gradient establishment time, and acceptable bacterial densities in the chemotaxis chamber (Supplementary Fig. 1). Thousands of trajectories were acquired in independent replicates. Sample sizes were deemed sufficient as they yielded reproducible and statistically significant results. Five biologically independent replicates in *E. coli* show similar results. Biologically independent experiments use cultures obtained from different overnight cultures. Figure 1b shows data for one replicate. For Fig. 1d, e, we combine data from three biological repeats each yielding three recordings. The profile of the drift velocity as function of z position is similar in each replicate (Supplementary Fig. 1g). Supplementary Table 1 details statistical characteristics of the datasets used in the analysis and related figures.

For *C. crescentus*, we combined data from five biologically independent experiments for analysis. Descriptors of the resulting dataset and individual subsets used for specific types of analysis are detailed in Supplementary Table 2. While a statistically significant drift velocity could only be computed on the combined data set, a chemotactic modulation of average run durations is evident for each individual experiment (Supplementary Fig. 3). A control dataset without a gradient shows similar run durations in opposite directions.

**Reporting Summary**. Further information on research design is available in the Nature Research Reporting Summary linked to this article.

## Data availability
Data shown in figures is provided as Supplementary Data 1. All trajectory data are available in the Harvard Dataverse repository at https://doi.org/10.7910/DVN/7DF0AT[37]. Any other data can be obtained from the corresponding author upon reasonable request.

## Code availability
Bacterial tracking was performed using a custom Matlab algorithm previously described[12].

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

## Acknowledgements

We thank H. Berg for providing *E. coli* strain AW405, and H. Berg, B.G. Hosu, S. Crosson, A. Fiebig, D. Hershey, N. Cira, A.W. Murray, and M. Burns for helpful comments on the manuscript. This research was funded by the Rowland Institute at Harvard.

## Author contributions

K.M.T. conceived the research, M.G. and K.M.T. designed the experiments, M.G. performed all experiments and analysis, M.G. and K.M.T. interpreted the data and wrote the manuscript.

## Competing interests

The authors declare no competing interests.
