## [Peer Review File · Communications Biology]

Reviewers' comments:

Reviewer #1 (Remarks to the Author):

This paper describes two sets of 3D tracking experiments on bacteria, one using *E. coli* and a second using *Caulobacter crescentus*. The authors make some interesting and (to my knowledge) novel observations about the qualitative difference between chemotaxis 2D and 3D experiments in *E. coli*, and develop a model for the switching behavior of *C. crescentus* in which the forward and reverse states have the same underlying energy, and are separated by a barrier with variable height. This contrasts with *E. coli*, in which the energies of the two states vary, leading to independently varying run/tumble durations.

I found the structure of this paper a little strange. In some ways it is two different papers, one on *E. coli* and one on *C. crescentus*, jammed together and linked quite superficially by the assay method. This makes following the narrative difficult at times, but the authors have made good use of figures which helps. I think that this study is worthy of publication, subject to the concerns that I list below being addressed.

Major issues

1) Are you convinced that the flick events exhibited by *C. crescentus* are changes in motor direction? When Xie and colleagues described the similar phenomenon in *Vibrio* (ref. 9), they note the opposite: the flagellum is behind the cell both before and after the 'flick'. This would complicate your analysis of the *Caulobacter* trajectories, in which you've described the flicks as motor reversals.

2) The manuscript quotes the number of 'trajectories' captured, but it's hard to convert this to the number of individual cells. The quote of ~80,000 trajectories seems difficult to square with statements about capturing up to 100 cells in each frame, and a movie of three minutes. Can the authors estimate how many distinct individuals are captured in each movie?

3) Could the authors clarify where the threshold between motile and non-motile cells is? In the methods they seem to say that the cut-off speed is either 10 $\mu\text{m/s}$ ('In-device conditions') or 15 $\mu\text{m/s}$ ('Data analysis'). Which is it and why? I note that Berg's original paper (ref 6) gave the average swimming speed of the strain that the authors use as 14.2 $\mu\text{m/s}$, so a threshold of 15 $\mu\text{m/s}$ is going to select for the highest-speed swimmers, and may not be truly representative of the population (assuming the same media etc. were used). The results of Supp. Fig. 1d seem to show a different average speed, but this could be sample-to-sample variation. Either way, it would be good to know.

4) The measured drift speed for *Caulobacter crescentus* is low, as the authors note. Is there a statistical way of assuring the reader that this isn't a false positive? The remarks about smooth swimmers in the supplementary material are an interesting nuance (smooth swimmers moving down the gradient).

Minor issues

P3, line 14 'the rate of surface encounters'. Presumably smooth swimmers are more likely to remain trapped at the boundary as well?

P4, line 4, *C. crescentus* shows little variability in motor bias between individuals – presumably the authors exclude the smooth swimmers from this statement?

p.12 ref. 20 – should be a space in between 'Crescentus' and 'obeys'

Lastly, the supplementary material contains a large amount of interesting and in-depth discussion of the main results, and the main manuscript's discussion is quite brief. If the journal's style guide and word limit allow, the authors could consider moving some of this material to the main part of the manuscript - I leave this at the authors' discretion though.

Reviewer #2 (Remarks to the Author):

In the paper from Grognot & Taute, the authors use a 3D bacterial tracking routine to study the chemotactic behavior of *E. coli* and *C. crescentus* in a 3D microfluidic assay. *E. coli*, being the paradigm to study chemotaxis, is used to validate the technique and discuss the impact of the 3D or 2D geometry on the quantification of the chemotactic performances, with a particular focus on the effect of the channel surfaces on the bacterial trajectories. The technique is then exploited to study the chemotactic performances of *C. crescentus*, which are not well characterized. The authors show that in the population of *C. crescentus* there are two different phenotypes: "smooth swimmers", which do not show appreciable chemotactic drift, and swarmer cells, which display a small chemotactic drift. The authors also show that *C. crescentus* performs chemotaxis with a constant motor bias mechanism, in contrast with the mechanism used by *E. coli*. The paper is an interesting and a timely contribution to the field, well suited to the journal readership. It is clearly written, the quality of the figure is high and the methods are accurately described, the data analysis procedure is accurate and well described and extended Supplementary Information is provided. I would only recommend to clarify two details regarding the estimate of the drift velocity of *C. crescentus*, mainly due its very small value, which could be easily affected by artifacts. The first one regards the comment of the authors on the possible effect of glucose (see comment 4) and the second one regards the negative drift measured in the smooth swimming part of the population of *C. crescentus* (see comment 5).

1. Line 17, page 1: In order to assess the statistics used to support the findings, could a more precise estimation of the number of trajectories collected in a representative experiment be given, as for *C. crescentus*?
2. Methods: For *E. coli*, the authors report the data for three replicates of the chemotaxis experiment, while for *C. crescentus* this information is not reported. Could the author explicitly mention how many experimental replicates were run and what are the data obtained in each one?
3. Methods: What is the impact of the stalked *C. crescentus* cells attached on the channels surface on the optical quality of the images? Could it affect the accuracy of the technique?
4. Line 1-3, Page 22: The information that the presence of glucose could interfere with xylose chemotaxis should be reported in the discussion on *C. crescentus* chemotactic performances in the manuscript, since it could possibly affect the interpretation of the results. Could a control experiment with glucose as chemoattractant be performed?
5. Line 6-11, page 23: I wonder if the imbalance in concentration due to the xylose was quantified and its magnitude could justify the entity of the drift. Did the authors consider possible strategies to compensate it? I understand the experimental difficulties addressing this point may entail, however, the impact of this effect on one of the main results of the paper (i.e. the measurement of the drift up the gradient for *C. crescentus*) is too relevant for it to be underrated. If this point cannot be addressed experimentally, it deserves to be reported and fully discussed in the manuscript.

Response to Reviewers
Grognot & Taute

We thank the reviewers for their positive evaluation. We are pleased that reviewer 1 finds our study “worthy of publication” and reviewer 2 considers our work “an interesting and a timely contribution to the field, well suited to the journal readership”.

Upon the reviewers’ thoughtful feedback, we have added a number of improvements to our manuscript. Most importantly, we now present additional analyses which verify that the small drift velocity measured for *C. crescentus* is not an artifact, as well as additional experimental data and theoretical analyses that confirm that the down-gradient drift observed for smooth swimmers is consistent with effective diffusion driven by an imbalance in bacterial concentrations. We now evaluate competition between chemotaxis and diffusive drift in a new section in the Results. The Discussion section has been expanded to comprise some material previously presented in the Supplementary Discussion, as requested by reviewer 1, as well as a new section evaluating the sensitivity limits of the technique. Results and Discussion are now separate sections. In addition, we have revised details for clarity and style throughout the text.

Below we address each comment specifically. Full reviewers’ comments are shown in blue, our responses in black, and quotations from the manuscript in grey. Edits to the manuscript text are shown as underlined additions or crossed out ~~deletions~~.

Best regards,
K.M. Taute

Referee expertise:

Referee #1: high speed digital holographic microscopy, biophysics of swimming microbes

Referee #2: optical techniques for flow visualisation

Reviewers' comments:

Reviewer #1 (Remarks to the Author):

This paper describes two sets of 3D tracking experiments on bacteria, one using *E. coli* and a second using *Caulobacter crescentus*. The authors make some interesting and (to my knowledge) novel observations about the qualitative difference between chemotaxis 2D and 3D experiments in *E. coli*, and develop a model for the switching behavior of *C. crescentus* in which the forward and reverse states have the same underlying energy, and are separated by a barrier with variable height. This contrasts with *E. coli*, in which the energies of the two states vary, leading to independently varying run/tumble durations.

I found the structure of this paper a little strange. In some ways it is two different papers, one on *E. coli* and one on *C. crescentus*, jammed together and linked quite superficially by the assay method. This makes following the narrative difficult at times, but the authors have made good use of figures which

helps. I think that this study is worthy of publication, subject to the concerns that I list below being addressed.

Major issues

1) Are you convinced that the flick events exhibited by *C. crescentus* are changes in motor direction? When Xie and colleagues described the similar phenomenon in *Vibrio* (ref. 9), they note the opposite: the flagellum is behind the cell both before and after the 'flick'. This would complicate your analysis of the *Caulobacter* trajectories, in which you've described the flicks as motor reversals.

The reviewer is correct that Xie et al (PNAS 2011) and also Son et al (Nat Phys 2013) show that there is a delay between the switch in flagellar rotation direction from pulling to pushing and the buckling-induced reorientation characteristic of a flick. This delay is very short however: Son et al. report 10 ± 5 ms for *V. alginolyticus*. At our recording rate of 30 Hz, the delay is not resolved, and the flick events we record encompass both the reversal accompanying the motor switch from pulling to pushing as well as the buckling-induced reorientation.

For practical purposes, we can thus consider flicks to coincide temporally with motor reversals.

2) The manuscript quotes the number of 'trajectories' captured, but it's hard to convert this to the number of individual cells. The quote of ~80,000 trajectories seems difficult to square with statements about capturing up to 100 cells in each frame, and a movie of three minutes. Can the authors estimate how many distinct individuals are captured in each movie?

The *C. crescentus* data encompass five biologically independent experiments, cumulating a total of 75 min of recording, as reported in the methods. We have now added a clarifying statement in the main text as well (p.3 ll.5-7):

We recorded 79,244 individual 3D bulk trajectories of motile *C. crescentus* cells navigating a 1 mM/mm xylose gradient in five biologically independent experiments (Supplementary Table 3).

Our ~123,000 s of analyzable trajectory time (motile with a minimum trajectory duration of 0.8 s) thus imply that each frame contains an average of about $123,000 \text{ s} / (75 * 60 \text{ s}) = 22$ individuals that meet the selection criteria. Other individuals may be present, but many trajectories are too short to meet the selection criterion, and a few are not motile. Note also that the cell density increases with time during each individual *C. crescentus* experiment, as the bacteria are grown in the sample chamber.

We now also evaluate the statistical data requirements for detecting a given drift velocity in a new section in the Supplementary Discussion (Section 3, p.31 l.25 – p.32 l.17) which we summarize in a section "Sensitivity of the technique" in the Discussion section of the main text (p.5 l.38 – p.6 l.12).

3) Could the authors clarify where the threshold between motile and non-motile cells is? In the methods they seem to say that the cut-off speed is either 10 $\mu\text{m/s}$ ('In-device conditions') or 15 $\mu\text{m/s}$ ('Data analysis'). Which is it and why? I note that Berg's original paper (ref 6) gave the average swimming speed of the strain that the authors use as 14.2 $\mu\text{m/s}$, so a threshold of 15 $\mu\text{m/s}$ is going to select for the highest-speed swimmers, and may not be truly representative of the population (assuming the same media etc. were used). The results of Supp. Fig. 1d seem to show a different average speed, but this could be sample-to-sample variation. Either way, it would be good to know.

Typically, distributions of average swimming speeds are bimodal, with one peak due to motile and one peak due to immotile cells. We have now added an example of such a speed distributions for *E. coli* to S. Fig. 1 as panel f, reproduced below. Because the motile swimming speeds and the apparent speed of non-motile cells depend on strains and conditions, we generally choose the threshold for considering cells motile as a value that separates those two peaks.

Supplementary Figure 1f) Distribution of individual average swimming speeds for the full population of bulk trajectories (red, weighted by trajectory duration) and of instantaneous swimming speeds for the motile population (blue), defined as having an average speed larger than a threshold (grey), set at 15 $\mu\text{m/s}$ for this experiment. Data shown are for the same experiment as for Fig. 1b.

Howard Berg's seminal 1972 paper indeed reports very low swimming speeds, while also noting that the 3D tracker was unable to follow the faster individuals in the population. More importantly, swimming speeds strongly depend on growth conditions. The 1972 study used growth in minimal medium with glycerol, a poor carbon source. Later work from the Berg lab that uses growth conditions similar to ours (TB at 33°C, Turner, Ryu & Berg, *J. Bact.* 2000) finds average swimming speeds of approximately 30 $\mu\text{m/s}$ for strain AW405, practically identical to those observed by us here.

The reason a lower motile speed threshold was chosen for the supplementary experiments testing in-device conditions in S. Fig. 1d-e is that, in these first experiments performed to test the assay, lower average swimming speeds were observed overall. We attribute this discrepancy to a learning curve in bacterial handling. For instance, flagella may break during washing steps, or cells may become de-energized during extended periods on the bench at room temperature. The chemotaxis experiments reported here were performed later, with more practice, and yield reproducible average swimming speeds of approximately 30 $\mu\text{m/s}$ as shown in Supplementary Table 2. We opted not to repeat the earlier experiments testing in-device conditions because the conclusions only rely on a lack of temporal changes in swimming speed, not the absolute values.

We have added information on average swimming speeds observed for all experiments in the newly added Supplementary Table 2.

4) The measured drift speed for *Caulobacter crescentus* is low, as the authors note. Is there a statistical way of assuring the reader that this isn't a false positive? The remarks about smooth swimmers in the supplementary material are an interesting nuance (smooth swimmers moving down the gradient).

We thank the reviewer for raising this issue. While indeed the drift velocity is so low that we cannot differentiate it from zero unless we combine datasets from multiple repeats, the modulation of the run duration as a function of direction relative to the gradient is apparent in the individual underlying datasets. We now show these data in Supplementary Figure 3, reproduced below. Importantly, a control

dataset, obtained in the absence of a gradient, shows no effect. Thus, the behavioral mechanism driving chemotaxis is apparent in several independent experiments. We thus trust that the measured drift velocity is indeed due to chemotaxis, although its value may underestimate the chemotactic drift due to the additional presence of diffusive drift. We have added a quantitative analysis evaluating the consistency of the observed drift with the observed behavioral modulation in the new section “Competition between chemotaxis and diffusion”.

Supplementary Figure 3: Reproducibility of *C. crescentus* chemotaxis experiments. a) Average run durations of runs leading up or down the gradient (defined as in Figure 2) in 5 biological replicates in the presence of a 1 mM/mm xylose gradient as well as one control experiment without a gradient, a) for all runs, b) for runs of identified orientation (backwards/forwards). c) Average durations of forward versus backward runs in each of the experiments in panels a and b, for runs leading up (yellow) or down (cyan) the gradient. Error bars represent standard errors of the means.

We have also undertaken additional experiments to quantify the xylose-driven bacterial concentration imbalance between the reservoirs (Supplementary Figure 4, reproduced below) and added a theoretical analysis predicting the resulting diffusive flux to the Supplementary Discussion (Section 2.3, p.29 I.5 - p.30 I.9). We find that the observed down-gradient drift of smooth swimmers agrees well with the predicted diffusive flux.

We also evaluate possible diffusive components lowering the drift velocity of turning bacteria. These aspects are now discussed in a new section “Competition between chemotaxis and diffusion” in the Results section of the main text (p.4 ll.13-40).

Supplementary Figure 4: Xylose-driven bacterial density imbalance. a) Average swimming speeds observed over time in the left and right reservoirs of the chemotaxis chamber, containing xylose/M2G and M2G, respectively. The grey box marks the time period during which trajectories are recorded in the gradient in the middle of the chamber during chemotaxis experiments. b) Bacterial densities observed in the reservoirs over time. In panels a and b, small points indicate individual 50-s recordings, and circles with error bars show the mean and standard deviation of three such recordings obtained in close temporal proximity. c) Bacterial density of solutions retrieved from the two reservoirs after 53 min. d) Individual optical density at 620 nm from microplate reader measurements in 96-well plate, after blank subtraction, for 20 wells with M2G or 20 wells with xylose/M2G. e) Doubling times, determined by a linear fit to the logarithm of the optical density data in panel d between OD 0.08 and OD 0.18 against time, indicate that the growth rate is approximately 11% higher in the presence of xylose. Error bars reflect 95% confidence intervals of average doubling times across 20 wells for each condition.

Minor issues

P3, line 14 ‘the rate of surface encounters’. Presumably smooth swimmers are more likely to remain trapped at the boundary as well?

We thank the reviewer for pointing out the ambiguity of our wording – indeed both the rate of bacteria arriving at the surface and the time spent swimming near the surface are expected to increase for smooth swimmers. We have changed the wording in the manuscript accordingly (p.3 ll.17-18):

Smooth swimming is likely to increase both the rate of surface encounters as well as the time spent swimming along the surface and ...

P4, line 4, *C. crescentus* shows little variability in motor bias between individuals – presumably the authors exclude the smooth swimmers from this statement?

The statement reflects a finding reported in Ref. 20. We have changed the wording to *C. crescentus* ~~shows~~ has been reported to show hardly any variability in motor bias between individuals²⁰

to reflect this unambiguously (p. 4 ll. 9-10).

In principle, it is possible for smooth swimmers to exhibit the same motor bias as turning cells. Their bias would manifest itself in the fraction of smooth swimmer that swim in pushing vs pulling mode. It is however also plausible that whatever biological factors that distinguish the smooth-swimming and turning phenotypes also alter the motor bias. Our data offer limited insight into this question, which we now evaluate in the Supplementary Discussion (Section 2.2, p.28 l.28 – p.29 l.3). We thus refrain from making any claims about motor bias in smooth-swimming individuals.

p.12 ref. 20 – should be a space in between ‘Crescentus’ and ‘obeys’

We thank the reviewer for spotting the error which we have now corrected.

Lastly, the supplementary material contains a large amount of interesting and in-depth discussion of the main results, and the main manuscript’s discussion is quite brief. If the journal’s style guide and word limit allow, the authors could consider moving some of this material to the main part of the manuscript - I leave this at the authors' discretion though.

We are glad that the reviewer appreciated the depth of the discussion. We have moved the section “Magnitude of the *C. crescentus* chemotaxis response” to the Discussion section in the main text (p.5 ll.2-36) and expanded our analysis of the possible effect of density imbalances to a new section “Competition between chemotaxis and diffusion” in the Results section (p.4 ll.13-40).

Reviewer #2 (Remarks to the Author):

In the paper from Grognot & Taute, the authors use a 3D bacterial tracking routine to study the chemotactic behavior of *E. coli* and *C. crescentus* in a 3D microfluidic assay. *E. coli*, being the paradigm to study chemotaxis, is used to validate the technique and discuss the impact of the 3D or 2D geometry on the quantification of the chemotactic performances, with a particular focus on the effect of the channel surfaces on the bacterial trajectories. The technique is then exploited to study the chemotactic performances of *C. crescentus*, which are not well characterized. The authors show that the in population of *C. crescentus* there are two different phenotypes: “smooth swimmers”, which do not show appreciable chemotactic drift, and swarmer cells, which display a small chemotactic drift. The authors also show that *C. crescentus* performs chemotaxis with a constant motor bias mechanism, in contrast with the mechanism used by *E. coli*.

The paper is an interesting and a timely contribution to the field, well suited to the journal readership. It is clearly written, the quality of the figure is high and the methods are accurately described, the data analysis procedure is accurate and well described and extended Supplementary Information is provided.

We thank the reviewer for their positive evaluation of our work.

I would only recommend to clarify two details regarding the estimate of the drift velocity of *C. crescentus*, mainly due its very small value, which could be easily affected by artifacts. The first one regards the comment of the authors on the possible effect of glucose (see comment 4) and the second one regards the negative drift measured in the smooth swimming part of the population of *C. crescentus* (see comment 5).

1. Line 17, page 1: In order to assess the statistics used to support the findings, could a more precise estimation of the number of trajectories collected in a representative experiment be given, as for *C. crescentus*?

We apologize for the oversight and have added a table (Supplementary Table 2) that lists statistical descriptors of the *E. coli* datasets. Furthermore, all trajectory data are now provided for download in the Harvard Dataverse repository at <https://doi.org/10.7910/DVN/7DF0AT>.

2. Methods: For *E. coli*, the authors report the data for three replicates of the chemotaxis experiment, while for *C. crescentus* this information is not reported. Could the author explicitly mention how many experimental replicates were run and what are the data obtained in each one?

We pool data from 5 biologically independent experiments, as reported in the methods. We have now added this information also to the main text for increased transparency (p.3 ll.5-7):

We recorded 79,244 individual 3D bulk trajectories of motile *C. crescentus* cells navigating a 1 mM/mm xylose gradient in five biologically independent experiments (Supplementary Table 3).

The drift velocity is too low to be confidently resolved in a single experiment. The chemotactic modulation of run durations by the orientation relative to the gradient, however, is apparent in the individual experiments, but not in a control experiment with no gradient. These data are now presented in Supplementary Figure 3, which is reproduced above in the response to reviewer 1.

We now also evaluate the statistical data requirements for detecting a given drift velocity in a new section in the Supplementary Discussion (Section 3, p.31 l.25 – p.32 l.17) which we summarize in a section “Sensitivity of the technique” in the discussion (p.5 l.38 – p.6 l.12).

3. Methods: What is the impact of the stalked *C. crescentus* cells attached on the channels surface on the optical quality of the images? Could it affect the accuracy of the technique?

The optical background caused by the stalked cells attached to the surface is stationary and thus easily removed by our background correction routine, leaving only a small increase (up to approximately 25%) in the variability of background pixel counts which has no obvious impact on the tracking accuracy. The stalked cells do however interact physically with swimming cells in their vicinity, thus we exclude trajectory segments within 13 μm of the surface for *C. crescentus*, whereas for *E. coli* we used a threshold of 10 μm .

4. Line 1-3, Page 22: The information that the presence of glucose could interfere with xylose chemotaxis should be reported in the discussion on *C. crescentus* chemotactic performances in the manuscript, since it could possibly affect the interpretation of the results. Could a control experiment with glucose as chemoattractant be performed?

Our statement that glucose could interfere with xylose chemotaxis intended to refer to the possibility that, if xylose and glucose were sensed by the same receptor, then the chemotactic response to a xylose gradient could be lowered by the presence of a constant background of glucose. Any set of experimental conditions likely contains a large number of biological factors that affect the magnitude of the chemotactic response - e.g., growth conditions affect chemotaxis protein expression (Li & Hazelbauer, J. Bact., 2004; Ni, ..., Sourjik, PNAS 2018).

The conclusions of our work, however, do not rely in any way on the exact quantitative value of the drift velocity. Our key finding for *C. crescentus* is that our assay enables us to a) detect that there is chemotaxis and b) reveal the underlying chemotactic mechanism, extending both forward and backward swimming segments when ascending a gradient, in contrast to the *E. coli*-like mechanism previously assumed to hold. This mechanism is expected to be independent of the magnitude of the chemotactic response.

In addition, the question of whether glucose interferes with xylose chemotaxis would not be easily resolved by a control experiment with *C. crescentus* in a glucose gradient. If chemotaxis to glucose was observed, it would still be unclear whether the chemotactic responses to glucose and xylose affect each other.

Furthermore, such an experiment would also involve technical challenges that complicate the interpretation of the results. Because glucose is the main carbon source supporting growth in the experiments, a glucose gradient would lead to extremely non-uniform growth and hence strong diffusive fluxes opposing chemotactic fluxes due to the resulting density gradient.

Thus, while technical issues prevent us of from resolving whether glucose interferes with xylose chemotaxis, we argue that this question is a minor point that does not affect the conclusions we draw from our work. For clarity and transparency, however, we now note this point also in the main text as suggested by the reviewer. The section “Magnitude of the *C. crescentus* chemotaxis response” has been revised and moved from the Supplementary Information to the Discussion in the main text (p.5 ll.2-36).

5. Line 6-11, page 23: I wonder if the imbalance in concentration due to the xylose was quantified and its magnitude could justify the entity of the drift. Did the authors consider possible strategies to compensate it? I understand the experimental difficulties addressing this point may entail, however, the impact of this effect on one of the main results of the paper (i.e. the measurement of the drift up the gradient for *C. crescentus*) is too relevant for it to be underrated. If this point cannot be addressed experimentally, it deserves to be reported and fully discussed in the manuscript.

We thank the reviewer for raising this point. We have since performed additional experiments to quantify the relative bacterial concentrations in the two reservoirs as well as a theoretical evaluation of the expected diffusive drift arising from it and confirm their consistency with the observed drift.

We find that

- a) as expected, there is a difference in bacterial density between the two reservoirs, with the xylose-containing reservoir showing a density 1.8-fold higher than the other one,
- b) independent growth measurements show an increased growth rate in the presence of xylose,
- c) the down-gradient drift velocity observed for smooth-swimming bacteria of $-(0.8 \pm 0.2) \mu\text{m/s}$ agrees well with a theoretical estimate based on the observed concentration imbalance and active diffusion ($-0.6 \mu\text{m/s}$),
- d) the up-gradient drift velocity observed for turning bacteria is consistent with a competition of up-gradient chemotaxis and down-gradient active diffusion.

Points a and b are shown in Supplementary Figure 4 which is reproduced above in the response to Reviewer 1.

The possible competition between chemotaxis and diffusion is now discussed in a full section in the main text (“Competition between chemotaxis and diffusion”, p.4 ll.13-40), and a detailed theoretical analysis is presented in the Supplementary Discussion (Section 2.3, p.29 l.5 – p.30 l.9).

For point d, we exploit the fact that we can use the measured dependence of the average run duration on the orientation relative to the gradient to produce an estimate of the magnitude of solely the chemotactic drift that is not affected by active diffusion. Coarsely speaking, the overall measured drift velocity arises from a combination of how runs up and down the gradient differ in duration (which is driven by chemotaxis) and how they differ in number (which is driven by active diffusion).

We emphasize again that the exact value of the drift velocity is not crucial to the conclusions of this work. Our key points are that we can determine that there is chemotaxis, even if weak, that we can determine the behavioral mechanism that underlies it, and that this behavioral mechanism differs from the *E. coli* scheme that so far has been assumed to apply to *C. crescentus*.

REVIEWERS' COMMENTS:

Reviewer #1 (Remarks to the Author):

I thank the authors for their responses to my previous comments; the original submission was an interesting and careful study, and their response has essentially allayed whatever remaining concerns that I had. Sample-to-sample variation bedevils experiments like this, but the authors have carefully controlled for these effects. There are a few results that are counterintuitive at first sight (e.g. the increased number of smooth swimmers in the xylose reservoir in Supp. Fig. 4c) but I'm grateful that the authors have taken the time to explain these.

The only outstanding issue that I have is that the authors haven't really answered my second question – can they estimate how many different motile cells are captured in total? I'm certainly not implying that their conclusions aren't sound (they capture a very large number of trajectories), but it might help other authors to establish the standard of proof in the field. For example, an upper limit might be $123,000 \text{ seconds} / 0.8 \text{ seconds} = 154,000 \text{ cells}$ – but this is taking the minimum, rather than the average trajectory duration. I expect that there might be cell trajectories that are captured multiple times, so perhaps such an estimate isn't straightforward. At any rate, I'm happy to leave this at the discretion of the authors/editors.

Reviewer #2 (Remarks to the Author):

This manuscript is concerned with obtaining further insights into bacterial motility and chemotaxis. For this purpose, authors have relied on advanced tracking techniques with microscopy to image multiple bacteria in 3D.

-The authors studied the gradient stability with fluorescein. However, methyl aspartate, which has about 1/3rd of molecular weight of fluorescein. In other words, methyl aspartate is expected to have a higher diffusivity. It is not certain that "gradient stability" with fluorescein ensures the same with methyl aspartate for the time-scale selected.

- The reviewer is concerned with the relative time scales of diffusion and mixing for bacteria and chemoattractant. Using simple calculations, one can assume a diffusion coefficient of $1 \times 10^{-9} \text{ m}^2/\text{s}$ for methyl aspartate. Within 10 minutes, the length scale that will be covered with diffusion is $(D \cdot t)^{1/2}$ is about 700 μm , which is larger than the length central zone being imaged (400 μm). Hence, the exact meaning and nature of gradient should be discussed and better analyzed.

-The time and spatial resolution of the method needs to be mentioned in the methods. Depending on the relative magnitude of resolution scale and the length scales of bacterial transport, different interpretations can be deduced. This information can be added to "Sensitivity of the technique" section.

-In addition, bacteria selected in this study has a rod-shape, which implies that non-isotropic diffusion in translation and rotational directions should be considered.

-When a bacterium approaches to the surface, the intermolecular forces will become significant below a critical distance. Namely, the bacteria-surface interaction will have a magnitude comparable with thermal energy, $k_B T$. At this point, the motion of bacteria near surface is dictated by the gradient of double-layer forces between these. In general, these forces are much more dominant over diffusion effects, usually below 50-200 nm (depending on the Debye length). The surface effects of motility must consider these aspects.

In terms of the big picture, the manuscript includes two main figures, summarizing the analysis of a large number of data/observations. The authors state that "the key advancement our assay is compatible with environments such as hydrogels." The questions are: is this manuscript proving information enough to advance the current-state-of-art in the area of bacterial motility? And have readers learned something new that has not been established in the literature? Considering many existing publications in this topic (e.g., <https://doi.org/10.1101/2020.08.10.244731> and <https://doi.org/10.1371/journal.pone.0217823>), the advancement in the field is unclear, which is needed for a top-tier journal publication. Overall, the reviewer suggests that the manuscript is to be transferred to Scientific Report rather than publishing in Nature- Communications Biology.

Response to reviewers

“A multiscale 3D chemotaxis assay reveals bacterial navigation mechanisms”

Marianne Grognot & Katja M. Taute

We thank the reviewers for their second assessment of our manuscript.

Below, we address individual comments in detail. Full reviewer comments are shown in blue, our responses in black, and manuscript excerpts in grey, with additions marked by underlining.

REVIEWERS' COMMENTS:

Reviewer #1 (Remarks to the Author):

I thank the authors for their responses to my previous comments; the original submission was an interesting and careful study, and their response has essentially allayed whatever remaining concerns that I had. Sample-to-sample variation bedevils experiments like this, but the authors have carefully controlled for these effects. There are a few results that are counterintuitive at first sight (e.g. the increased number of smooth swimmers in the xylose reservoir in Supp. Fig. 4c) but I'm grateful that the authors have taken the time to explain these.

We are glad that we were able to resolve the reviewer's concerns to their satisfaction.

The only outstanding issue that I have is that the authors haven't really answered my second question – can they estimate how many different motile cells are captured in total? I'm certainly not implying that their conclusions aren't sound (they capture a very large number of trajectories), but it might help other authors to establish the standard of proof in the field. For example, an upper limit might be $123,000 \text{ seconds} / 0.8 \text{ seconds} = 154,000 \text{ cells}$ – but this is taking the minimum, rather than the average trajectory duration. I expect that there might be cell trajectories that are captured multiple times, so perhaps such an estimate isn't straightforward. At any rate, I'm happy to leave this at the discretion of the authors/editors.

We apologize if we have misunderstood the reviewer's concern regarding this point. The absolute numbers of trajectories as well as average trajectory durations are reported in the Supplementary Tables 1 and 2.

Indeed, it is likely that not all trajectories belong to distinct individual cells as bacteria are free to leave and re-enter the tracking volume. Additionally, a tracked individual may also be lost by the algorithm and found again later. Thus, one individual can, in principle, contribute multiple trajectories. This point is now stated in the Methods (p. 10) as follows:

One individual may, in principle, contribute more than one trajectory, either because it leaves and re-enters the tracking volume or because the tracking algorithm briefly loses it and then finds it again.

It is hard to estimate how many trajectories an individual contributes on average. Computer simulations could help to resolve the point, but we refrain from undertaking such an effort since the answer has little bearing on our findings. Averages such as swimming speeds and drift

velocities are weighted by trajectory duration, such that the weight given to information from one individual only depends on the total time it was tracked, and not on how many trajectories contribute to that time.

Only if the total number of individuals sampled is very low and differences between individuals are very large, individuals contributing multiple trajectories might lead us to underestimate errors due to correlations between trajectories from the same individual. Our jackknifing procedure, which resamples data by selecting random sets of trajectories, however, produces error estimates very similar to variations between experiments. This indicates that we are far from a regime where a lack of sampling of different individuals has an impact on the data.

Reviewer #2 (Remarks to the Author):

This manuscript is concerned with obtaining further insights into bacterial motility and chemotaxis. For this purpose, authors have relied on advanced tracking techniques with microscopy to image multiple bacteria in 3D.

-The authors studied the gradient stability with fluorescein. However, methyl aspartate, which has about 1/3rd of molecular weight of fluorescein. In other words, methyl aspartate is expected to have a higher diffusivity. It is not certain that “gradient stability” with fluorescein ensures the same with methyl aspartate for the time-scale selected.

The molecular mass of methyl aspartate is about half that of fluorescein. Molecular weight, however, is not a good indicator of binary diffusivity on the scale of small molecules.

Experimental data, however, are available. We now address this point in the Methods (p. 9):

Diffusive timescales scale with the inverse of the diffusion coefficient, and the diffusion coefficient of fluorescein³⁵ is about half that of aspartate³⁶. We assume that the diffusion coefficient of methyl aspartate would be in a similar range as that of aspartate. An aspartate gradient would thus establish even more rapidly than a fluorescein gradient and be stable about half as long as a fluorescein gradient. The stability of a fluorescein gradient over 4.5 h thus implies the stability of an aspartate gradient over 2 h. Our measurements are conducted within 1 h, well within the limits of stability of the chemical gradient.

- The reviewer is concerned with the relative time scales of diffusion and mixing for bacteria and chemoattractant. Using simple calculations, one can assume a diffusion coefficient of $1 \cdot 10^{-9} \text{ m}^2/\text{s}$ for methyl aspartate. Within 10 minutes, the length scale that will be covered with diffusion is $(D \cdot t)^{1/2}$ is about 700 μm , which is larger than the length central zone being imaged (400 μm). Hence, the exact meaning and nature of gradient should be discussed and better analyzed.

We are uncertain what the reviewer might be referring to with “mixing” as opposed to diffusion. The chemical gradient in the channel is established by diffusion over timescales t set by the diffusion coefficient $D \approx 10^{-9} \text{ m}^2/\text{s}$ and the length of the channel ($d = 1\text{mm}$), $t = d^2 / (2D) =$

500 s. Consistent with this estimate, we observe that a gradient is formed within minutes. The length scale of the imaged region does not play into these considerations.

Over very long time scales, diffusion would equilibrate the chemical concentrations of the two reservoirs. This time scale is much larger than the time scale of our experiments, as argued above.

Perhaps the reviewer is concerned whether bacteria migrating up the gradient might lead to an imbalance in bacterial concentrations? We have now added the following point to the Methods (p. 9):

The chemotactic drift of bacteria up the gradient is not sufficient to cause a substantial imbalance in bacterial concentrations between the two reservoirs. Assuming a uniform drift speed of approximately 3 $\mu\text{m/s}$ along the 1 mm long channel, approximately $3600 \text{ s} \times 3 \mu\text{m/s} / (1000 \mu\text{m}) \approx 11$ channel volumes of bacteria are transported up the gradient in one hour. The volume of the channel is $V_c = 1 \text{ mm} \times 1 \text{ mm} \times 0.1 \text{ mm} = 0.1 \mu\text{l}$, much smaller than the 65 μl volumes of the reservoirs. In one hour, the bacterial concentration of the reservoirs thus would change by at most $(11 * 0.1 \mu\text{l}) / 65 \mu\text{l} \approx 1.6\%$ as a result of chemotactic drift.

-The time and spatial resolution of the method needs to be mentioned in the methods. Depending on the relative magnitude of resolution scale and the length scales of bacterial transport, different interpretations can be deduced. This information can be added to "Sensitivity of the technique" section.

While it is generically true that time and spatial scales have an impact on experimental readouts, we are not sure where the reviewer sees ambiguities in the interpretation of our results. Data acquisition rates and time scales are reported; and so are the spatial scales of the assay. Thus, we are not sure what different interpretations specifically the reviewer has in mind.

-In addition, bacteria selected in this study has a rod-shape, which implies that non-isotropic diffusion in translation and rotational directions should be considered.

Translational Brownian diffusion is negligible in most bacterial motility studies.

Approximating a bacterial cell as a sphere of diameter $d = 2 \mu\text{m}$, its translation Brownian diffusion coefficient is $D_b = k_B T / (3\pi d \eta) \approx 2.2 \cdot 10^{-13} \text{ m}^2/\text{s}$, where $\eta \approx 10^{-3} \text{ N s/m}^2$ is the viscosity of water. Its displacement vt due swimming with speed v becomes larger than that due to diffusion, $\sqrt{2D_b t}$, for times larger than $t = 2D_b / v^2$. With $v = 30 \mu\text{m/s}$, we obtain $t = 0.5 \text{ ms}$, more than an order of magnitude shorter than our temporal resolution, and several orders of magnitude smaller than the time scales of bacterial behavior modulation (typical run durations are in the range of 0.2 – 2 s for many species). The corresponding displacement is about 15 nm, two orders of magnitude smaller than our spatial resolution.

Rotational diffusion does play a role, in particular in the straightness of runs. This has been discussed in the literature, starting with Berg & Brown, Nature 1972. It does, however, not have a bearing on the results reported here.

-When a bacterium approaches to the surface, the intermolecular forces will become significant below a critical distance. Namely, the bacteria-surface interaction will have a magnitude comparable with thermal energy, $k_B T$. At this point, the motion of bacteria near surface is dictated by the gradient of double-layer forces between these. In general, these forces are much more dominant over diffusion effects, usually below 50-200 nm (depending on the Debye length). The surface effects of motility must consider these aspects.

The attraction of motile bacteria to surfaces is quantitatively well described by solely hydrodynamic effects, see ref 30, indicating that intermolecular forces are negligible by comparison. In addition, displacements due to active motility are typically dominant over diffusive effects, as argued above, thus the relative strengths of diffusive effects and intermolecular forces is likely not an important factor here.

Most importantly, however, our manuscript makes a statement about the effect of the observed surface interactions on chemotaxis, not their causes. The causes of the surface interactions are beyond the scope of manuscript and have already been analyzed in detail elsewhere (aside from ref. 30, see other work by Eric Lauga and references therein).

In terms of the big picture, the manuscript includes two main figures, summarizing the analysis of a large number of data/observations. The authors state that “the key advancement our assay is compatible with environments such as hydrogels.”

We were unable to identify the source of the quotation above. It appears to be neither from our manuscript, nor from our rebuttal letter. The statement is, however, correct in content. The assay is currently used in our lab to investigate chemotaxis in complex environments such as hydrogels. The closing sentence of our manuscript (reproduced below, from pp. 6-7) provides an outlook towards future publications reporting such experimental results. To avoid any potential for ambiguity, we have added an “also” so as to reflect that are assay can be used in both liquid and complex environments:

In contrast to many flow-based chemotaxis assays that are limited to liquid environments, our assay is also compatible with environments such as hydrogels that more closely mimic the complexities of many natural habitats, and thus paves the way for studies of chemotactic mechanisms in ecologically relevant settings.

The questions are: is this manuscript proving information enough to advance the current-state-of-art in the area of bacterial motility? And have readers learned something new that has not been established in the literature? Considering many existing publications in this topic (e.g., <https://doi.org/10.1101/2020.08.10.244731> and <https://doi.org/10.1371/journal.pone.0217823>), the advancement in the field is unclear, which is needed for a top-tier journal publication. Overall, the reviewer suggests that the manuscript is to be transferred to Scientific Report rather than publishing in Nature- Communications Biology.

The substantial number of publications on bacterial motility and chemotaxis indeed indicates that it is an exciting field relevant to a large scientific community.

Our manuscript presents not only the first use of high-throughput 3D tracking for chemotaxis experiments but uses its 3D advantage to reveal previously unrecognized weaknesses of typical 2D chemotaxis assays as well as to uncover that *E. coli*'s motility strategy does not translate broadly to other species, contrary to widely held beliefs. Both findings are highly relevant to researchers in the field as they impact the design and interpretation of other experiments.